# Stability and Reactivity of Guaiacylglycerol-*β*-Guaiacyl Ether, a Compound Modeling *β-O*-4 Linkage in Lignin

Zeinab Rabiei [1], Andrew Simons [1], Magdalena Folkmanova [2], Tereza Vesela [2], Ondrej Uhlik [2], Evguenii Kozliak [1,*] and Alena Kubátová [1,*]

[1] Chemistry Department, University of North Dakota, Grand Forks, ND 58201, USA; rabieizeynab@gmail.com (Z.R.); andrew.j.simons@und.edu (A.S.)

[2] Department of Biochemistry and Microbiology, University of Chemistry and Technology, Prague, Technicka 3, 16628 Prague, Czech Republic

[*] Correspondence: evguenii.kozliak@und.edu (E.K.); alena.kubatova@und.edu (A.K.); Tel.: +1-701-7770348 (A.K.)

**Abstract:** Lignin, a complex and abundant biopolymer, is a major constituent of plant cell walls. Due to its chemical and structural complexity, lignin degradation is a challenging task for both natural and engineered systems. Therefore, investigation of lignin degradation using so called "model compounds" has been the focus of many research efforts in recent years. This study addresses the utility of guaiacylglycerol-*β*-guaiacyl ether (Gβ2) as a model compound for evaluating the *β-O*-4 bond cleavage under diverse thermal and aqueous medium conditions. Experimental conditions included varied pH (3–10), microbial biodegradation, subcritical water environment (150–250 °C), and mild pyrolysis (150–250 °C). A high-performance liquid chromatography with high-resolution mass spectrometry was employed for accurate detection and quantification of both Gβ2 and its degradation/modification products in an aqueous environment. Pyrolysis experiments were performed using gas chromatography-mass spectrometry analysis with a pyrolyzer. The results showed that Gβ2 remained stable under exposure to moderate pH and several bacterial strains, which were successfully used previously for biodegradation of other recalcitrant pollutants. We report, for the first time, differing Gβ2 breakdown pathways for subcritical water treatment vs. pyrolysis under an inert atmosphere. The scientific novelty lies in the presentation of differences in the degradation pathways of Gβ2 during subcritical water treatment compared to pyrolysis in an inert atmosphere, with water playing a key role. The observed differences are ascribed to the suppression of homolytic reactions by water as a solvent.

**Keywords:** guaiacylglycerol-*β*-guaiacyl ether; lignin model compound; subcritical water; pyrolysis; biodegradation

## 1. Introduction

Lignocellulosic biomass is a biocomposite material consisting mainly of cellulose, hemicellulose, and lignin. Lignin, as one of the main biomass constituents (15–35%) responsible for the rigidity of plant cell walls, is an important renewable feedstock for production of phenolic monomers, which are viewed as a possible replacement of fossil fuels [1].

Natural lignin is a heteropolymer consisting primarily of three phenylpropanoid monomers including *p*-hydroxyphenyl (H), guaiacyl (G), and syringyl units (S) (Scheme 1a) [2]. However, unlike regular biopolymers, e.g., proteins or carbohydrates, these common structural moieties are interconnected by a number of different strong linkages (Table 1) yielding an irregular and rather recalcitrant structure [3]. As a result, lignin typically depolymerizes into a mixture of phenolic derivatives (shown in Scheme 1b) occurring as both monomers and oligomers. Such monomers, e.g., those shown in Scheme 1b, are different from lignin

structural monomers shown in Scheme 1a. For example, guaiacol derivatives are formed out of the guaiacyl unit [4]. These products, i.e., monoaromatic phenolic derivatives, are also called "phenolic monomers" to distinguish them from other, less useful products of lignin degradation containing several linked aromatic moieties, e.g., dimers, trimers, and higher-order oligomers [5].

**Scheme 1.** Chemical structures of phenolic monomers: (**a**). monolignols, the actual monomers of lignin synthesis and (**b**). examples of monoaromatic phenolic products of lignin degradation.

When targeting this feedstock's ultimate utilization, its structural complexity requires the use of representative model compounds reflecting specific motifs of lignin structure to effectively study lignin degradation pathways. These model compounds are usually phenolic dimers, which convert into a defined mixture of specific monomers as a result of breaking targeted linkages.

In this study, we focus on the most abundant linkage, the alkyl aryl ether $\beta$-*O*-4 bond (Scheme 2), which represents more than 50% of all linkages that exist in native softwood lignin [6]. A number of other, less abundant linkages occur as well, and some corresponding model compounds were also proposed and used in previous studies. However, those less abundant connections are outside of the scope of this particular study.

**Scheme 2.** Chemical structures of: (**a**) general model compound featuring a *β*-*O*-4 linkage and (**b**) specific model compound, guaiacylglycerol-*β*-guaiacyl ether (G*β*2) used in this study.

One of the most recognized and used model compounds featuring both the guaiacyl moiety and *β*-*O*-4 linkage is guaiacylglycerol-*β*-guaiacyl ether (Scheme 2), which is often abbreviated as Gβ2, although the alternate GGGE acronym is also used. Other model compounds with this linkage are less representative because they do not possess the guaiacyl moiety. This dissimilarity with the actual lignin structural features may affect both their reactivity and solubility in water [7,8]. A variety of approaches to lignin breakdown were applied using these model compounds including oxidative [9], reductive [10], catalytic [11], and biological degradation [12]. Those studies performed in aqueous media with or without catalysts are summarized in Table 1.

The analysis of this literature (Table 1), however, reveals two significant knowledge gaps pertaining to the use of lignin model compounds. First, while a variety of product analysis methods were applied, only a few studies addressed the analysis of both reactant (e.g., Gβ2) and products using a *single* protocol [12–14]. Namely, most of the previous studies employed gas chromatography-mass spectrometry (GC-MS) for product characterization. This method enables a reliable identification and quantification of volatile reaction products, i.e., guaiacols and other methoxyphenol derivatives [14]. However, the direct GC-MS analysis does not allow for the simultaneous determination of reactants, i.e., hydroxylated dimer model compounds, most of which are either not GC-elutable, as is Gβ2, or have low thermal stability [12–15]. The quantification was thus restricted to the breakdown products in most of the conducted studies (Table 1), thus limiting the options to conduct mass balance closure.

The above-mentioned papers that used a single GC-MS method for determination of both Gβ2 and its decomposition products employed the derivatization of hydroxyl groups [12,13]. However, this approach is not ideal for quantification, as the derivatization protocol must be optimized for any specific condition to assure its completion. Another approach is to quantify products directly by high-performance liquid chromatography with high resolution or tandem mass spectrometry (HPLC-HR MS or MS-MS) as reported by Rahimi et al. [14].

The second current knowledge gap is the shortage of baseline *β*-*O*-4 degradation data under moderate conditions in aqueous environments. While investigating various process conditions for model compounds' hydrolytic breakdown in water, the previous studies either used multiple additives (catalysts, co-catalysts, medium components, etc.), or conducted thermal breakdown under rather drastic conditions. In particular, various studies investigated the stability of lignin model compounds in heated aqueous media in the presence of catalysts (Table 1). Yet, little consideration was given to their stability in subcritical water (under sufficient pressure to maintain a liquid state) as a solvent.

Nagel et al. reported a Gβ2 hydrothermal decomposition in water without additives as well as in aqueous media containing varying concentrations of hydroxide and carbonate bases [13]. However, considering the reaction volumes used, the reactions investigated in that study appeared to occur mainly in the gas phase (i.e., steam) rather than in liquid subcritical water, which may not be relevant as a baseline to many applications. Furthermore, the reactivities of steam and liquid water can be expected to differ, as the dielectric constant and thus the polarity are significantly reduced for steam [16]. On the other hand, the other studies referenced in Table 1 used water in its liquid phase. However, they also employed many different additives, which made it difficult to separate the influence of redox, thermal, and acid-base factors.

As for the biodegradation occurring in aqueous media under ambient conditions, the impact of pH and blank mineral medium has not been assessed. To the best of our knowledge, only two studies evaluated the biodegradation of lignin model compounds, both claiming significant Gβ2 conversion rates [12,17], although without detailed quantitative data [12]. The impact of biodegradation was evaluated in this study using three different strains. Of those, *Burkholderia xenovorans* LB400 is an example of aerobic bacteria that activate bond cleavage in polychlorinated biphenyls (PCBs) [18]. It has been shown that the extracellular media of two other aromatic hydrocarbon degrading microorganisms, *Pseudomonas putida* JAB1, and *Rhodococcus jostii* RHA1, enable the lignin breakdown occurring due to the genes encoding for the extracellular dye-decolorizing peroxidases (DyP), dypA and dypB [19,20].

**Table 1.** Literature data on model compounds featuring the *β-O-4* linkage and summary of their degradation data in aqueous media including the products obtained and the analysis protocols.

| Model Compound | Depolymerization Method & Conditions | Breakdown Products | Analysis &Quantification Approach | Ref. |
|---|---|---|---|---|
|  Guaiacylglycerol-*β*-guaiacyl ether (Gβ2) | **Hydrothermal decomposition under neutral and basic conditions**. Model compound (MC) (50 mg) in $H_2O$ (3 mL) with/without a base (NaOH, $Na_2CO_3$), $N_2$, 175 °C for 15 min. |  | GC-MS internal standard (IS) based quantification of silylated products including Gβ2 using N,O-bis(trimethylsilyl)trifluoroacetamide (BSTFA) quantified using standards and effective carbon number (ECN) factors, closing mass balance. | [13] |
|  2-Phenylethyl phenyl ether | **Catalytic cleavage in the aqueous phase**. MC (1.98 g), $Ni/SiO_2$ (0.30 g), $H_2O$ (80 mL), 6 bar $H_2$, 120 °C, 90 min, stirring at 700 rpm. |  | GC-MS with IS based quantification, using 2-isopropylphenol. Specific conditions of carbon mass balance determination NR. [a] | [21] |

**Table 1.** *Cont.*

| Model Compound | Depolymerization Method & Conditions | Breakdown Products | Analysis &Quantification Approach | Ref. |
|---|---|---|---|---|
| Gβ2 | **Aqueous phase hydrodeoxygenation**. MC (100 μg), solid acid zeolite, $H^+$-Y (300 mg), $Ru/Al_2O_3$ (300 mg), $H_2O$ (30 mL), 40 bar $H_2$, 250 °C, 4 h |  | GC-MS with quantification, using *n*-decane as an IS. The quantification appears to be based only on normalization. Gβ2 conversion NR. [a] | [22] |
| 2-Phenoxy-1-phenylethanol | **Base promoted hydrogenolysis over metal catalysts in water**. MC (10 mol%); $Ni_7Au_3$, NaOH, $H_2O$, 10 bar $H_2$, 130 °C, 0.5 h. | <br>Plus 12 minor products: 7 dimers and 8 monomers | GC-MS identification, GC-FID with quantification using standards and effective carbon number (ECN) factors. *n*-Dodecane was used as an IS. | [23] |
| Gβ2 | **Catalysis with $H_2$ donor agent (reduction); deoxygenation pathways under $H_2$ transfer conditions**. MC (0.067 g/L) in 50% EtOH in $H_2O$, Pt/C (5.12 g), 80 bar, 275 °C. |  | GC-MS identification, quantification approach NR. | [24] |
| Gβ2 | **Catalytic hydrogenolysis. MC (0.5 mmol)**. 50% THF in $H_2O$ (20 mL), Fe-L1/C-800 catalyst (0.1 g), 10 bar $H_2$, 240 °C, 12 h. |  | GC-MS with IS based quantification of products using phenol, dodecane to determine toluene. Gβ2 conversion NR | [11] |
| 2-Phenoxy-1-phenylethanol | **Catalytic redox-neutral C–O bond cleavage**. MC (0.1 mmol), Pd on metal-organic framework, EtOH/$H_2O$ at 120 °C, 6 h under $N_2$. Pressure value NR. [a] |  | GC-MS quantification, the quantification approach NR, Gβ2 conversion NR. [a] | [25] |
| Gβ2 and its oxidized derivative | **Formic acid-induced depolymerization**. MCs (1 mmol), sodium formate (3 mmol), $HCO_2H:H_2O$ (10:1) (5 mL), 110 °C, 12 h. Pressure value NR. |  | Identification and quantification by LC-MS, LC-MS-MS and GC-MS with/without derivatization. Both products and feedstock MCs were quantified. | [14] |

**Table 1.** *Cont.*

| Model Compound | Depolymerization Method & Conditions | Breakdown Products | Analysis &Quantification Approach | Ref. |
|---|---|---|---|---|
| 2-Phenoxyacetophenone | **Bioconversion with a Gram-negative bacterium,** *Acinetobacter* sp. TUS-SO1 cells with 0.1 g/L yeast extract and 1% (*v/v*) Tween 80 with MC (1 mM) in DMSO, 1% [*v/v*] at 30 °C for 72 h, shaking |  | HPLC-MS quantification of product and MC feedstock. GC-MS with trimethylsilylation confirmed identification. | [17] |
| Gβ2 | **Bioconversion** (*Burkholderia* sp. ISTR5). MC (2,000 mg/L), with overnight grown bacteria to obtain a final optical density of 0.1 at 30 °C for 72 h shaking. |  | GC-MS with BSTFA derivatization; identification of both products and MC, quantification approach NR. | [12] |

[a] NR denotes not reported.

The present study was designed to investigate the Gβ2 stability toward both abiotic hydrolysis at moderate pH and biodegradation, as well as its thermal degradation at moderate temperatures, with and without subcritical water, using a reverse phase HPLC with high-resolution time-of-flight mass spectrometry (TOF-MS) for analysis. The results obtained in the aqueous environment were compared to breakdown pathways via mild thermolysis under inert atmosphere using pyrolysis (Pyr)-GC-MS. Therefore, we set out to obtain a "baseline" for any future studies to be conducted with this model compound when using various catalysts and more severe process conditions, ultimately leading to the development of new and improved methods for lignin utilization. As a result of using this method, this study enabled the evaluation of the impact of water as a solvent on the Gβ2 reactivity and decomposition pathways, by detecting and identifying the reaction products including any side reactions.

## 2. Materials and Methods

### 2.1. Materials

Water was obtained from Millipore Direct-Q UV, and LC-MS grade acetonitrile (ACN) was purchased from Fisher Scientific. Gβ2, used in biodegradation, abiotic hydrolysis, and subcritical water experiments, was synthesized at UND [26]. For pyrolytic experiments, this substrate was replaced with pure Gβ2 purchased from Sigma Aldrich St. Louis, MO, USA. All other standards including vanillin, 4-chloroacetophenone (used as an internal standard, IS), guaiacol, and 4-hydroxy-3-methoxycinnamaldehyde were purchased from Sigma Aldrich (see their structures in Figure S1). Three bacterial strains, *Paraburkholderia xenovorans* LB400, *Pseudomonas alcaliphila* JAB1, and *Rhodococcus jostii* RHA1 were tested for Gβ2 degradation. *Paraburkholderia xenovorans* LB400 was purchased from American Type Culture Collection, *Pseudomonas alcaliphila* JAB1 and *Rhodococcus jostii* RHA1 were obtained from the collection of strains in the Department of Biochemistry and Microbiology, UCT Prague. Bacteria were cultured in a mineral salt solution (MSS, described elsewhere [27]) with biphenyl (Sigma Aldrich) and sodium pyruvate (Sigma Aldrich) added, along with either yeast extract (Sigma Aldrich) or Gβ2.

### 2.2. Assessment of Gβ2 Stability

For **pH effect studies**, Gβ2 aqueous solutions (300 µg/mL) were prepared at three different pH values with acetic acid (0.1 M pH 2.9 rounded to 3 henceforth), ammonium acetate (0.1 M, pH 7.3 rounded to 7 henceforth), and ammonium hydroxide (0.1 M, pH 10.3 rounded to 10 henceforth), in two different container materials, i.e., glass and plastic vials. The Gβ2 stability was evaluated after exposure to these media for 0–7 days.

In **biodegradation experiments**, we followed the procedure used in our earlier study [27], with the essential modifications due to the use of Gβ2 as a substrate. The optimum substrate concentration and harvesting time-points were determined as follows. Bacterial cultures growing in MSS medium with sodium pyruvate and yeast extract (30 mM and 0.001%, respectively) at an exponential phase (optical density at 600 nm, $OD_{600}$, of 0.5–1.0) were centrifuged at $5000 \times g$ for 10 min and washed twice with MSS. Cells were resuspended in MSS to a final $OD_{600}$ of 0.025. Autoclaved cell suspensions (121 °C for 20 min) were used as controls. Three replicates of 1 mL active or autoclaved cells or MSS were incubated with 0.1, 0.3, 0.5, 1.0, 3.0, 5.0, or 10.0 mM Gβ2 in 8 mL glass vials sealed with screw caps at 28 °C on a rotary shaker at 130 rpm. Every 24 h, the $OD_{600}$ of the culture was measured in microvolumes (50 µL). According to the measured growth curves, the optimal concentration of Gβ2 with which the culture reached the maximum $OD_{600}$ (5.0 mM) was determined. This concentration was then used during the assays described below.

The harvesting time points corresponding to the exponential phase, maximum $OD_{600}$ and stationary phase were determined; these time points were then used during the degradation assays (DGA), which were conducted as follows. Three replicas of 1 mL active or autoclaved cells or MSS were incubated with 5.0 mM Gβ2 in 8 mL glass vials sealed with screw caps for 0, 20, 40, and 110 h at 28 °C on a rotary shaker at 130 rpm. Destructively harvested samples were stored at −20 °C.

Biodegradation of Gβ2 was assessed through a resting cell assay (RCA) using the following protocol. All strains were at least thrice re-inoculated in fresh medium with biphenyl as a sole carbon source to support the expression of degradative genes. Bacterial cultures growing in MSS with 5.0 mM biphenyl as a sole carbon source were filtered when reaching the exponential phase ($OD_{600}$ of 0.5–1.0) with a coffee filter to remove biphenyl crystals, centrifuged at $5000 \times g$ for 10 min and washed twice with MSS. Cells were resuspended in MSS to a final $OD_{600}$ of 0.5. Autoclaved cell suspensions (121 °C for 20 min) were used as controls. Four replicas of 1 mL active or autoclaved cells were incubated with 5 mM/1600 ppm Gβ2 in 8 mL glass vials sealed with screw caps for 0, 24, 48, and 72 h at 28 °C on a rotary shaker at 130 rpm. Destructively harvested cultures in vials were stored at −20 °C until analysis.

For analysis, a sample aliquot (0.1 mL) was transferred to an Eppendorf tube and extracted with 60% MeOH in water (0.5 mL). The extraction was completed using vortexing and refrigeration for 15 and 10 min, respectively. Then, an aliquot of the extract (0.1 mL) was filtered using a syringe filter and 0.9 mL of water and IS were added.

The **subcritical water experiments** were conducted in a lab scale batch reactor previously described [28]. Briefly, the system consisted of a GC oven (Hewlett-Packard GC5890) and rotor made of a Leeson permanent magnet DC gear motor (Grafton, WI, USA) equipped with a Dayton DC speed control. The rotary part was implemented in the thermally insulated door of the oven. The heated part of the rotor contained five holders to stir Parker vessels rated to 517 bars (Cleveland, OH, USA) at an approximate rate of 3 rpm to provide sufficient mixing.

An aqueous solution of Gβ2 without additives was prepared using 0.7 µL of Gβ2 (final concentration 280 µg/mL) and water with a total volume of ~2.4 mL. The experiment was set up in a way that ensured sufficient headspace while maintaining the pressure required to ensure that water was present in subcritical state and **safe operation (preventing overpressurization)**, according to water/steam equilibrium tables (Table 2), similar to our earlier work [28]. The specific experimental details are included in the supplementary information. The total reaction time was one hour. Three different temperatures, 150, 200,

and 250 °C, were applied. Five replicates of each experiment were performed with the variance shown as standard deviation.

**Table 2.** Saturation conditions for water [29].

| Reaction temperature (°C) | 150 | 200 | 250 |
|---|---|---|---|
| Water saturation pressure (bar) | 4.780 | 15.55 | 39.76 |
| Liquid phase density (g/cm$^3$) | 0.9167 | 0.8647 | 0.7989 |
| Vapor density (g/cm$^3$) | 0.002545 | 0.007861 | 0.01997 |

For **mild pyrolysis experiments**, Gβ2 was used as a solid. The procedure was similar to that used in earlier studies [28]. Stainless steel Eco-cups were cleaned using a butane blowtorch until glowing hot to remove any impurities that could be carried over from previous experiments. The cups were allowed to cool to room temperature. Each analyte was individually placed in a prepared Eco-cup and weighed (30–60 µg) using a microbalance. Following the introduction of the analyte, a precleaned quartz filter (heated at 600 °C for 24 h) was placed on top of each sample in each Eco-cup.

All experiments were conducted in triplicate unless otherwise stated. The data (concentrations and/or degradation %) were reported as mean values ± one standard deviation. The *t*-test with 95% confidence was used to assess the statistical significance of the obtained values compared to the blanks/controls.

### 2.3. HPLC-TOF-MS and Pyr-GC-MS Analyses

The analyses, except for the pyrolysis experiment, were performed using a reverse phase HPLC Agilent 1100 Series system equipped with a diode array detector (DAD) and a TOF-MS detector, Agilent G1969A (Agilent Technologies, Santa Clara, CA, USA), with an electrospray ionization (ESI) source. The target analytes were separated using a gradient elution program in 30% ACN in water as described below. The separations were performed using an Agilent Zorbax Eclipse Plus C18, 2.1 × 159 mm, 3.5-µm column. The injection volume was 20 µL. The mobile phase consisted of 2.5 mM formic acid in water (solvent A) and 2.5 mM formic acid in acetonitrile (solvent B). The binary gradient program started with 0 min, 30% B (0–0.1 min), followed by a linear gradient to 90% B (0.1–10 min), then 90% B (10–13 min). The last step was conducted from 13 to 16 min with a gradient to 30% B followed by an equilibration 7 min hold preparing the system for the next analysis. The total run time was 23 min. The ESI was performed (after the optimization described in Table S1) in the positive ion mode, with a capillary voltage, 5000 V; fragmentor voltage, 125 V; nebulizer pressure (nitrogen), 25 psig; drying gas (N$_2$) flow rate, 12 L min$^{-1}$; gas temperature, 350 °C. The HPLC-TOF-MS data were recorded across the mass range of 50–1000 *m/z*.

The results were processed with Agilent MassHunter software version 10.0. Based on the obtained HPLC-TOF-MS data, i.e., peak retention times and mass spectra, and characteristic ions, Gβ2 and all products were identified and quantified (Supporting Figures S1 and S2). The quantification was done using standards when available. For the rest of the analytes, the response factors of vanillin and Gβ2 were used for quantification of monomers and dimers, respectively. For the quantitative data processing, the representative ions of target analytes were extracted within the ±0.03 amu range (Table 3).

The pyrolysis experiments were performed using online **pyrolysis with GC-MS.** The instrument consisted of a Frontier 3030D pyrolyzer with an autosampler connected to an Agilent 7890 GC instrument with an Agilent 5975C MS detector having an electron ionization source. The pyrolysis was performed at selected temperatures (150, 200, 250 °C) for 30 s. The GC inlet temperature was set at 300 °C and the analysis was operated in 1:20 split mode with a helium flow rate of 1 mL/min. The GC oven temperature was held constant at 300 °C, the column was 25 m long with 0.25 mm I.D. and a 0.25 µm stationary phase composed of 5% phenyl polydimethylsiloxane (Ultra ALLOY$^+$-5, Frontier). The MS

analysis was carried out without a solvent delay in the 10–650 *m/z* range. Based on the obtained Pyr-GC-MS data, i.e., peak retention times and mass spectra, all products were identified either by the NIST library or pure chromatographic standards (HMC, G, and V) (Table 4).

**Table 3.** HPLC-TOF-MS analysis of Gβ2 and products of its hydrotreatment breakdown: retention times and the ions used for identification of the corresponding compounds.

| Compound Name, Abbreviation | Retention Time, min | Monoisotopic Mass, *m/z* | Quantification Ion [M + H]⁺ *m/z* | Confirmation Ion [M + Na]⁺ *m/z* | ID & Quant. Confirmation * |
|---|---|---|---|---|---|
| Guaiacylglycerol-β-guaiacyl ether, Gβ2 | 7.1 | 320.1260 | 321.1333 | 343.1152 | Standard |
| Vanillin, V | 5.9 | 152.0473 | 153.0546 | 175.0366 | Standard |
| Guaiacol, G | 6.9 | 124.0524 | 125.0597 | 147.0416 | Standard |
| 4-Hydroxy-3-methoxy cinnamaldehyde, HMC | 7.8 | 178.0630 | 179.0703 | 201.0522 | Standard |
| 1-(2′-Methoxyphenoxy)-2-(4′-hydroxy-3′-methoxyphenyl)ethane, D1 | 8.4 | 274.1205 | 275.1278 | 297.1097 | Tentative |
| 3-Hydroxy-2-phenoxy-1-(4′-hydroxy-3′-methoxyphenyl)-1-propanone, D2 | 8.9 | 288.0998 | 289.1071 | 311.0890 | Tentative |
| 2-(2′-Methoxyphenoxy)-3-(4′-hydroxy-3′-methoxyphenyl)propanal, D3 | 9.5 | 302.1154 | 303.1227 | 325.1046 | Tentative |

* The identification was either confirmed by standards or if unavailable, the tentative identification was based on high-resolution TOF-MS spectra with mass accuracy < 10 ppm. The tentatively identified dimers were quantified using the response factor for Gβ2.

**Table 4.** Retention times and molecular ions of compounds identified in the Pyr-GC-MS experiments.

| Compound Name, Abbreviation | Retention Time (min) | Molecular Ion *m/z* | Identification Type * |
|---|---|---|---|
| Vanillin, V | 6.7 | 152 | Standard |
| Guaiacol, G | 4.9 | 124.1 | Standard |
| 2-(2′-Methoxyphenoxy)-3-(4′-hydroxy-3′-methoxyphenyl)propanol, D3 | 11.3 | 302.1 | Tentative |
| *Cis/trans*-1,2-Di(4′-hydroxy-3′-methoxyphenyl) ethene, D4 | 11.6, 11.8, 12.8 for isomers | 272.1 | Tentative |
| 4-Hydroxy-3-methoxy cinnamaldehyde | 7.6 | 178.1 | Standard |
| Coniferyl alcohol, CA | 9.3 | 180.1 | Tentative |
| Homovanillin, H | 6.3 | 166.1 | Tentative |

* The identification was either confirmed by standards or, if unavailable, the tentative identification was based on the MS NIST 2020 library.

## 3. Results

### 3.1. Stability at Different pH

The first step was the assessment of Gβ2 stability under the conditions relevant to biodegradation, i.e., at room temperature and non-extreme pH values. The pH dependence of Gβ2 concentration before and after room-temperature hydrolysis is shown in Figure 1. The results obtained in both glass and plastic vials yielded similar data, thus indicating insignificant loss of target species due to sorption. The findings demonstrated that pH variation has only a minor impact on Gβ2 in terms of hydrolysis. The results obtained after

seven days, in both glass and plastic containers at pH 3 and 7, showed statistically insignificant changes, if any. However, in glass containers at pH 10, we observed a statistically significant decrease in the Gβ2 amount, which suggests an influence of adsorption in basic media on Gβ2 hydrolysis. One may conclude that Gβ2 is not prone to room-temperature hydrolysis under both acidic and neutral conditions.

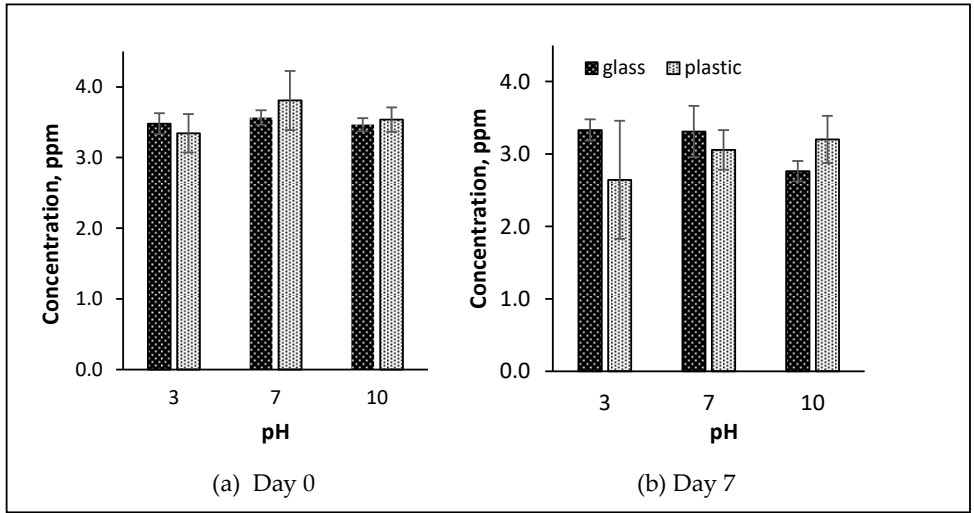

**Figure 1.** Effect of pH on Gβ2 concentrations in aqueous solutions with two container materials: (**a**) initial concentration and (**b**) concentrations after 7 days. The initial concentrations of Gβ2 used for experiments were ~300 ppm, the data shown reflect the dilution for analysis. The results are presented based on triplicates ± one standard deviation.

### 3.2. Evaluation of Gβ2 Biodegradation

Gβ2 biodegradation was investigated for three different bacterial strains previously shown to break down aromatic pollutants [18]. No statistically significant degradation was observed for JAB1 after 72 h of incubation. Under similar conditions with LB400 and RHA1, the Gβ2 amount decreased by ~20% compared to the control with dead cells (Figure 2), showing a minor, yet statistically significant effect.

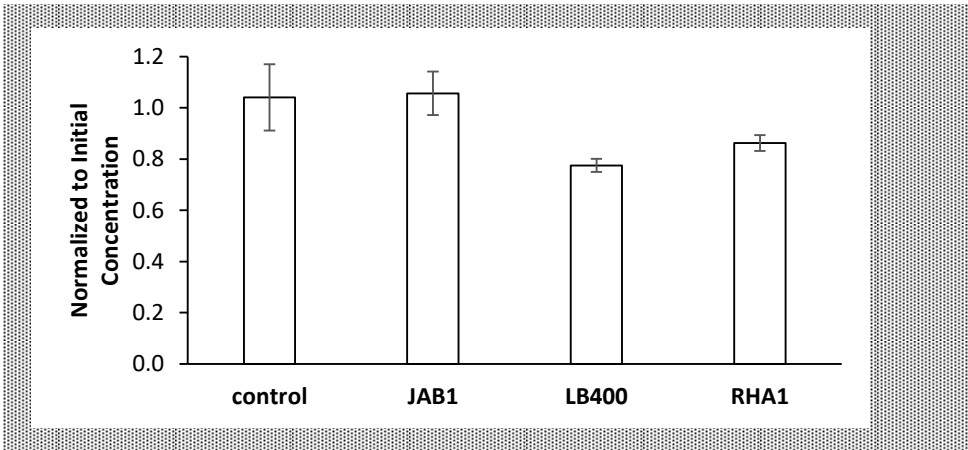

**Figure 2.** Gβ2 amounts in media with live biomass, taken as the ratios to those with dead biomass obtained under similar conditions at the same time, following average of 24–72 h incubation with control (medium with dead cells), *Pseudomonas putida* JAB1 strain, *Burkholderia xenovorans* LB400 strain, and *Rhodococcus jostii* RHA1 strain.

The loss of Gβ2 occurred within the first 24 h, and Gβ2 concentrations did not decrease further. Thus, despite extensive optimization of growth curves for all strains and

evaluation using RCA and DGA assays, no degradation products were observed or ongoing biodegradation. This is likely due to the recalcitrance of ether bonds in Gβ2. Other studies using *Burkholderia* sp. ISTR5 (R5) strain claimed successful lignin breakdown [12], yet a closer look at these published results shows that they were only qualitative, reporting the qualitatively assessed feedstock loss and products obtained by GC-MS following derivatization—but without quantification. Thus, the extent of biodegradation is difficult to determine, considering that response factors are different for the targeted analytes.

However, from the standpoint of creating a baseline for biodegradation experiments, the specific goal of this study has been fulfilled. Namely, no abiotic degradation was observed due to wall or medium effects, or adsorption to inactive biomass. This conclusion was confirmed by the lack of pH influence discussed in the previous section. Gβ2 proved to be more recalcitrant toward biodegradation than anticipated. Therefore, it appears to be suitable as a model substrate to assess lignin biodegradation.

### 3.3. Stability in Subcritical Water

In contrast to the observed stability of Gβ2 at moderate pH or under biodegradation conditions, its significant breakdown was observed in subcritical water at moderate temperatures of 150, 200, and 250 °C (Figure 3).

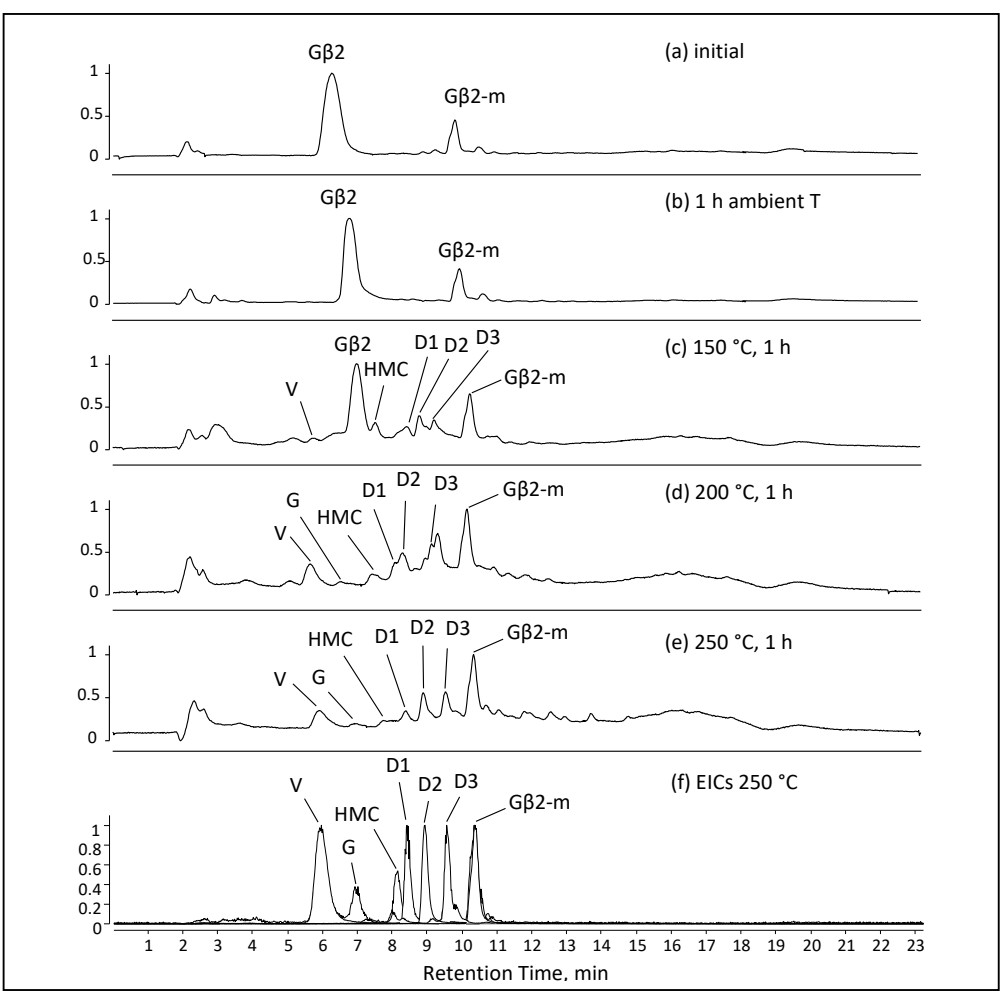

**Figure 3.** HPLC-TOF-MS total ion current chromatograms (TIC) (**a**–**f**) of (**a**) the Gβ2 sample before any treatment, (**b**) Gβ2 kept in the reaction vessel in water for 1 h at ambient conditions, then treated at (**c**) 150 °C, (**d**) 200 °C, and (**e**) 250 °C in subcritical water. (**f**) Extracted ion chromatograms (EIC) of the Gβ2 sample treated at 250 °C. For abbreviations of peak labels and extracted ions, see Table 3. Gβ2-m is an impurity in the synthesized Gβ2.

The HPLC-TOF-MS results demonstrated that Gβ2 undergoes a sizable conversion, even at 150 °C, while at 200 and 250 °C its conversion was complete. The same degradation products were observed at all three temperatures tested, but with different abundance. An extracted ion chromatogram confirming the identification of those products is provided as panel (f) of Figure 3.

As seen in Figure 3, HMC produced from Gβ2 decomposition was detected at all three temperatures, signifying that the $C_\beta$–O bond was cleaved. Furthermore, G (which is the other product of the target bond cleavage) was also present among products, confirming that the $C_\beta$–O bond is cleaved during the decomposition process. However, the other observed products, D1, D2, and D3, retained the $C_\beta$–O bond. In line with previous studies, V was also identified as a significant product [14].

Based on the observed molecular ions as protonated or sodium adducts (Table 3) detected with high-resolution TOF, we proposed the breakdown pathways shown in Scheme 3. Three different pathways appear to take place. First, there is the cleavage of a C–C bond, which resulted in producing V. Only traces of corresponding fragment were observed. Second, there is the cleavage of the targeted C–O ether bond, that resulted in G and HMC production. Several modified Gβ2 products formed with bond cleavages only in side chains, i.e., dimers, D1–D3, were also observed.

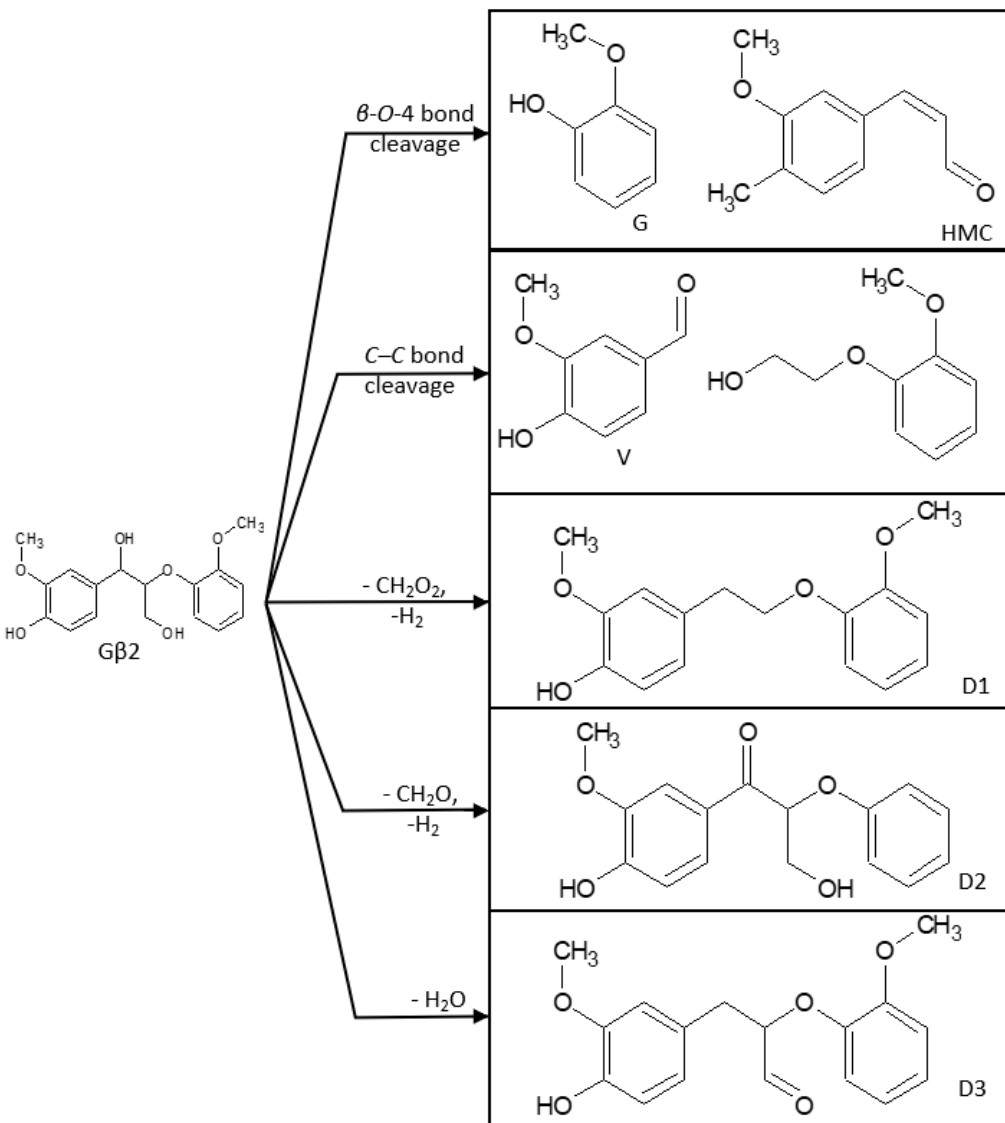

**Scheme 3.** Proposed breakdown products and pathway following the Gβ2 treatment in subcritical water.

Reactant and product quantification was performed as described in the Section 2 to close mass balance. Figure 4 shows the determined amounts of both Gβ2 and its thermal degradation products. Gβ2 showed some stability at 150 °C whereas it was virtually gone at 200 and 250 °C. Conversely, the amounts of guaiacol (G) and vanillin (V) produced increased with temperature, as expected, assuming that higher temperature promotes decomposition reactions. However, unexpectedly, these two products of ether bond cleavage represented only two minor reaction pathways.

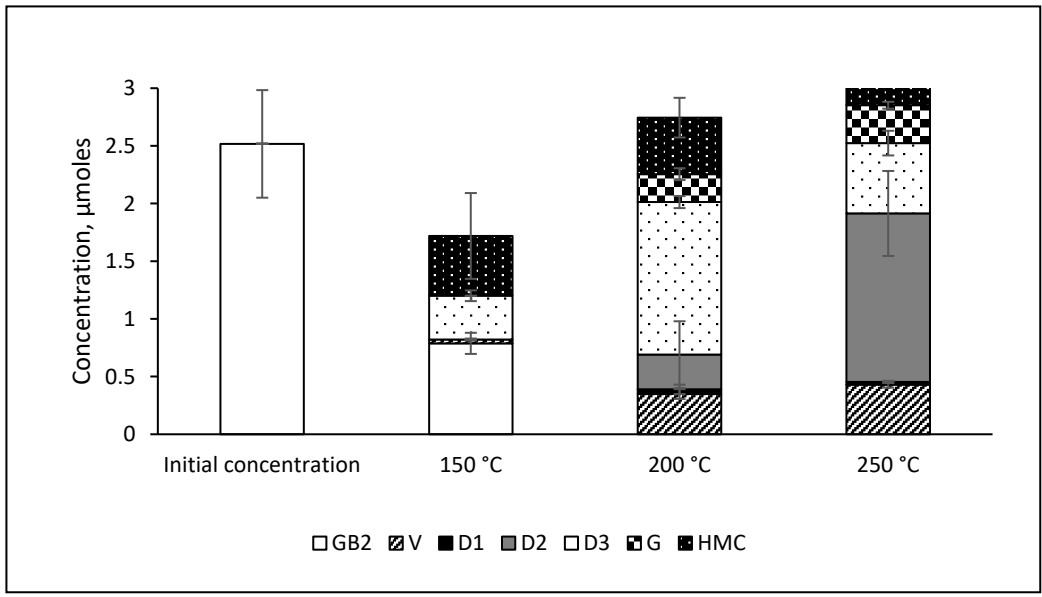

**Figure 4.** Reactant (Gβ2) and product amounts following a 1 h treatment in subcritical water (150, 200, and 300 °C. The results are presented based on triplicates ± one standard deviation.

The formation of dimer products (D2–D3), which was found to be the major reaction pathway, indicates the occurrence of significant side reactions that modify the Gβ2 structure in subcritical water without affecting the *β-O-*4 bond. D2 was more abundant at higher temperatures, whereas D3 formation showed a maximum at 200 °C. Thus, this chemical appears to be less stable than D2. Bond dissociation enthalpy assessment indicated that the C–O bond strength of the *β-O-*4 linkage is approximately 70 kcal·mol$^{-1}$ in both the native lignin and its common model compounds, and the presence of various substituents has merely a minimal impact on this value, within 3 kcal·mol$^{-1}$ [30]. However, this increment may be sufficient to shift the minimum degradation temperature, as in the case of D2 vs. D3.

One other factor appears to be important in effecting the varied *β-O-*4 bond stability in different compounds. The low Gβ2 conversion with ether bond scission in subcritical water reported in this study contrasts with much higher values observed with water at much lower pressure, i.e., steam [13]. Nagel and Zhang observed similar products with the intact *β-O-*4 bond—but only in minor amounts [13]. This difference indicates that Gβ2 is significantly more stable toward high temperatures in a polar solvent, i.e., subcritical water.

Presumably, a polar solvent tends to suppress homolytic reactions, while the heterolytic ether bond cleavage requires a greater energy. Consistent with this conclusion, Dou et al. reported that an altered reaction pathway for zeolite-catalyzed cleavage was observed when water was added to the ethanol solution [30]. It is of note that Gβ2 modification products with intact *β-O-*4 bonds appear to be characteristic for aqueous media as a liquid-phase solvent. One needs to be aware of this feature when using Gβ2 as a lignin model compound at high temperatures.

### 3.4. Stability toward Mild Solvent-Free Pyrolysis

To confirm the assumption that subcritical water protects Gβ2 from homolytic cleavage, a pyrolysis experiment was conducted without the presence of water in an inert helium atmosphere, using Pyr-GC-MS.

Figure 5 shows the Pyr-GC-MS TIC chromatograms obtained at three temperatures, 150, 200, and 250 °C. This information complements the other study, in which the pyrolysis of Gβ2 was investigated at much higher temperatures, 450, 550, 650, and 750 °C [31]. In contrast to our study, no dimers were detected in those high severity conditions, being replaced with catechols resulting from the scission of the other ether bond, within the methoxy group of Gβ2 [31]. The other study that investigated this process did not find any products but guaiacol at 200 and even 300 °C. Perhaps, the use of transfer lines in the Pyr-GC-MS setup precluded the detection of dimers and other less volatile products [32].

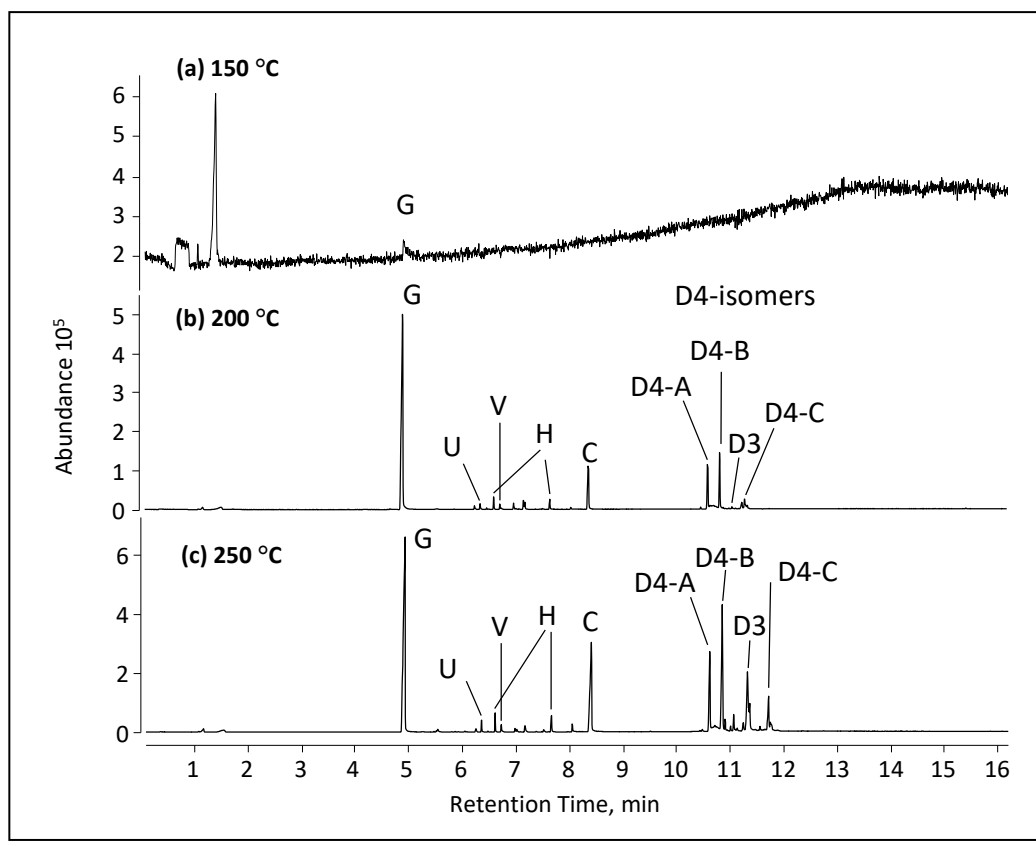

**Figure 5.** Pyr-GC-MS TIC chromatograms of the pure Gβ2 samples following the moderate pyrolysis at (**a**) 150 °C, (**b**) 200 °C and (**c**) 250 °C under inert helium atmosphere.

At moderate thermal conditions, G was the main product (Figure 5), indicating a smooth ether bond cleavage at 200 and 250 °C. This cleavage was presumably homolytic, as no medium of high polarity, i.e., water, was added. However, besides the main product, guaiacol, the obtained chromatograms showed a different product composition compared to the subcritical water treatment. Namely, the abundant peaks of dimers maintaining the *β-O-4* bond, D1–D3, were not observed. Only D3 was recovered in small amounts. In contrast, a different dimer, D4, was formed, occurring as three isomers. This dimer is proposed to be an alkene derivative, presumably formed as a result of a bimolecular reaction between guaiacol and coniferyl alcohol (or its derivatives) as shown in Scheme 4. Most likely, a guaiacyl radical is formed as an intermediate of guaiacol formation, which may attach itself to the second product at several positions. Thus, given the different functional groups observed in the dimer products, our conclusion on the significant solvent effect of water was confirmed.

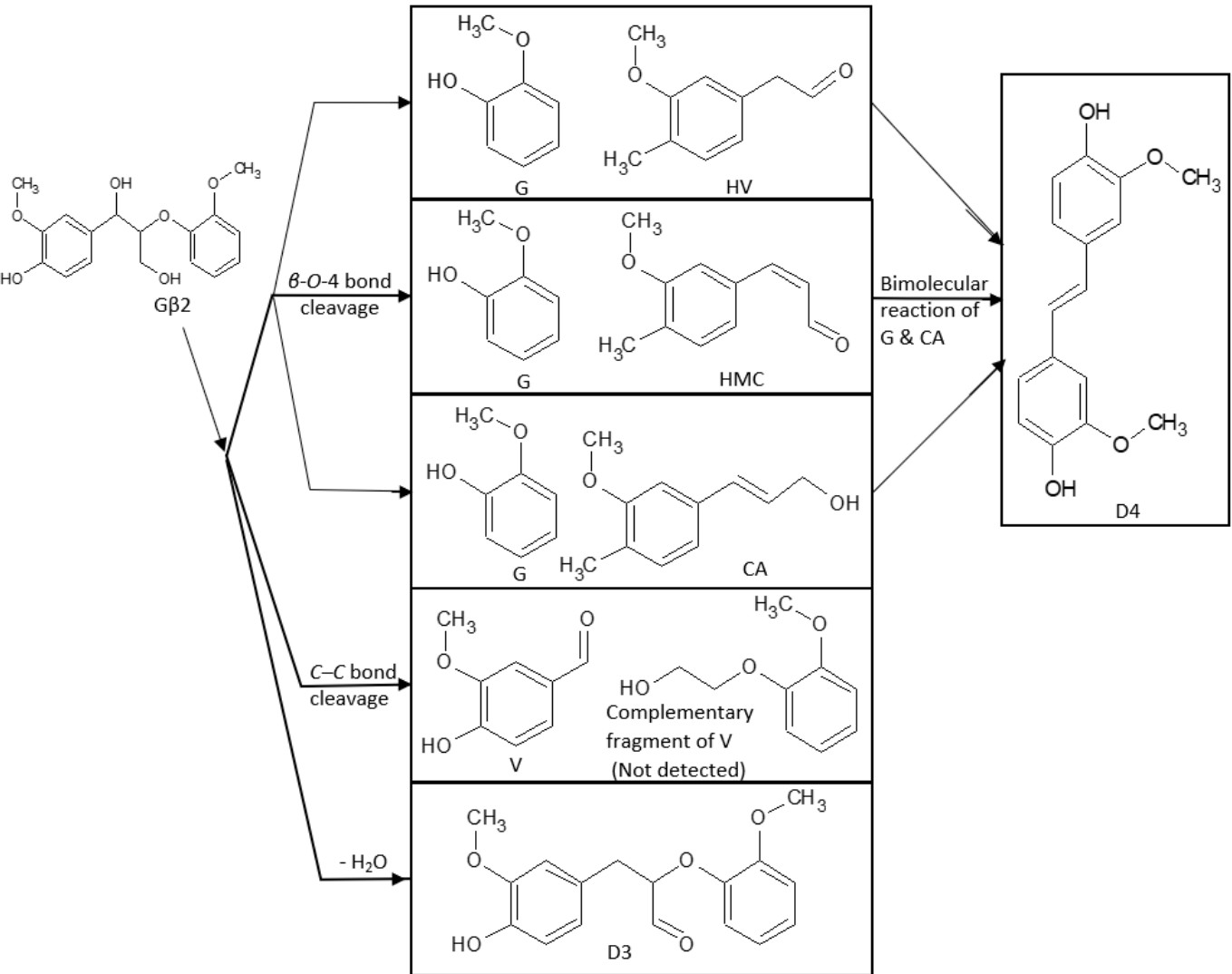

**Scheme 4.** Proposed Gβ2 moderate temperature pyrolysis breakdown pathways.

Furthermore, the guaiacol abundance proved to be greater than expected. In addition to G, the second product of the *β-O-4* bond cleavage, CA, was formed. However, this second product was formed in smaller amounts than G, even when the other monomer derivatives of CA, e.g., HMC, were taken into account. The observed excess yield of G was apparently due to its additional formation caused by the CC bond cleavage (presumably, homolytic) occurring on the other side of the Gβ2 molecule, as shown in Scheme 4. Low selectivity in bond breaking is characteristic for homolytic reactions, so this phenomenon should be expected. However, the indistinguishability of guaiacol molecules formed by the cleavage of two different bonds is an undesired feature for studies targeting specifically a *β-O-4* bond scission. This issue could be potentially addressed by using isotopically labeled model compounds, yet this was beyond the scope of this study. One has to beware of this issue when using model compounds with *β-O-4* bonds in non-polar solvents or solvent-free pyrolytic reactions.

## 4. Conclusions

This study investigated the efficiency of *β-O-4* bond scission in moderate conditions using guaiacylglycerol-*β*-guaiacyl ether (Gβ2) as the model compound. It appears to be suitable for biodegradation studies, because no artificial abiotic Gβ2 cleavage was observed. In the case of solvent-free pyrolysis, one has to be aware that guaiacol is formed from Gβ2, not only due to the *β-O-4* bond scission, but also as a result of C–C bond

cleavage on the other site of its molecule. However, this study showed only a limited Gβ2 applicability for thermal hydrolytic *β-O*-4 bond scission at 150–250 °C (i.e., in subcritical water), due to the occurrence of significant side reactions yielding other phenolic dimers, Gβ2 derivatives featuring much higher *β-O*-4 bond stability. The occurrence of such side reactions was shown to be due to the solvent effect of water protecting the reactant from homolytic thermal breakdown. The pathways of Gβ2 thermal degradation with and without subcritical water as a solvent were rather different, presumably reflecting the difference between heterolytic and homolytic bond cleavage, respectively. Extrapolating these results to lignin, the pathways of lignin depolymerization in subcritical water and as a result of solvent-free pyrolysis appear to have a different nature, leading to different products. The observed formation of the deoxygenated dimer as a result of Gβ2 pyrolysis (as opposed to subcritical water treatment) indicates that repolymerization of intermediates of lignin breakdown in that system may occur more readily.

**Supplementary Materials:** The following supporting information can be downloaded at: https: //www.mdpi.com/article/10.3390/separations11020059/s1, Figure S1: Structures of the compounds considered in this study; Figure S2: Mass spectra of the compounds used or observed in this study. Text: Experimental details for subcritical water treatment experiments. Table S1: Optimization of SI conditions with acquired responses for target ions dissolved in various solvent and electrolyte systems in positive/negative mode and varying fragmentor and capillary voltages performed using direct infusion.

**Author Contributions:** Conceptualization, A.K. and E.K.; methodology, Z.R.; software, Z.R.; validation, Z.R.; investigation, Z.R., M.F., T.V. and A.S.; data curation, O.U. and A.K.; resources M.F. and T.V.; writing—original draft preparation, Z.R.; writing—review and editing, A.K. and E.K.; project administration, A.K.; funding acquisition, A.K. and O.U. All authors have read and agreed to the published version of the manuscript.

**Funding:** The collaboration was enabled through National Science Foundation (NSF) IRES award # 1658615. The project was supported through U.S. Department of Energy's (DOE) Office of Energy Efficiency and Renewable Energy (EERE) Award DE-EE0009257. T.V. and M.F. were supported through ERASMUS, A.S. was supported through NSF S-STEM award #1742269. M.F., T.V., and O.U. received funding from the Czech Science Foundation (CSF) under grant no. 20-00291S. Any opinions, findings, conclusions, or recommendations expressed in this material are those of the author(s) and do not necessarily reflect the views of the NSF, DOE, or CSF.

**Data Availability Statement:** The raw data supporting the conclusions of this article will be made available by the authors on request.

**Acknowledgments:** We would like to express our gratitude to Irina Smoliakova for consultation on degrada tion pathways. Furthermore, we would like to acknowledge Nafisa Bala for operation and support with Pyr-GC-MS, and Chirby Ambo for HPLC work on biodegradation products.

**Conflicts of Interest:** The authors declare no conflicts of interest.

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
