# Peer review of "Stability and Reactivity of Guaiacylglycerol-β-Guaiacyl Ether, a Compound Modeling β-O-4 Linkage in Lignin"

_separations, doi:10.3390/separations11020059_

Round 1

Reviewer 1 Report

Comments and Suggestions for Authors

In the following manuscript from Kozliak, Kubatova et al, the degradation and analytical investigation of one of most used b-O-4 model compound is described. The authors investigated 4 differents degradation pathways, namely room temperature pH resistance, bio-digestion, hydrolysis, and pyrolysis and analyzed the results base on HPLC and GC techniques. The novelty of this work is to establish a single HPLC methodology to enable the investigation of diverse processes. However, HPLC methods have already been already largely proposed in litterature for either model compounds either wood either technical lignins ( eg 10.1039/D0EE02870Chttps://doi.org/10.1093/chromsci/bmaa105-10.1007/978-3-642-74065-7). The analytical results are farily poor, with low mass ballances, not resolved HPLC peaks and overall, the methods that has no demonstrated advantages over GCs or other already established methodology. Moreover, when taken separately, all the four investigations are conducted in an approximative  way, with conclusive remarks for complex reactions (bio-degradation, hydrolysis and pyrolysis) drawn using only one analytic techniques.   Therefore my reccomandation is the rejection of the manuscript. 

Author Response

We are resubmitting our manuscript, addressing the comments of all the Reviewers. We would like to thank all reviewers for their feedback, as it helped us to improve all manuscripts and more clearly define the purpose of the manuscript, which was perhaps not fully transparent in our initial version. 

Response:

First, we would like to address the concern about the lack of the novelty, as it is apparent that Reviewer 1 did not see the point of our study. This, perhaps, might suggest that we did not convey our efforts effectively. Yet, considering that the other three reviewers (as described below) did not have that view, we don’t think this is true. Nevertheless, we have given careful consideration to each of the concerns in this text below. We have also adjusted the text in the abstract to minimize any potential misunderstanding.

First, Reviewer 1 expressed a concern with the novelty of the work stating “The novelty of this work is to establish a single HPLC methodology to enable the investigation of diverse processes.” We disagree with this statement, and our point of view is supported by the other reviewers. Rather than addressing such a general objective (i.e., optimization of HPLC in applications to lignin chemistry), we set a specific goal of applying this method to investigate one concrete model system, with specific substrate and products. The developed separation method and product analysis enabled this application, leading to significant novel results. As Reviewer 4 pointed out, “The scientific novelty of the article lies in the presentation of differences in the degradation pathways of Gβ2 during subcritical water treatment compared to pyrolysis in an inert atmosphere, with water playing a key role.” Both Reviewer 2 and Reviewer 3 affirmed this notion. Specifically, Reviewer 3 stated that “It was found that there were differences in Gβ2 breakdown under subcritical water treatment vs. pyrolysis under inert atmosphere, which were ascribed to the suppression of homolytic reactions when water was used as solvent.” Echoing this statement, Reviewer 4 adds, “The experimental plan presented by the authors is quite bold and unpredictable, but the arguments they provide allow us to accept the conclusions drawn that the model substance Gβ2 retains its stability both when exposed to moderate pH levels, and several bacterial strains.”

We have revised the abstract to ensure that the novelty is conveyed more effectively.

As for the concern about HPLC  reflected in the statement of Reviewer 1 “HPLC methods have already been already largely proposed in literature for either model compounds either wood either technical lignins,” HPLC indeed has been applied for general lignin decomposition product analysis for decades. Yet these developments, particularly in the area of LC-MS, were carried out in much more recent works than the relatively outdated 1992 reference provided by the reviewer. It is of note, though, that they were not used as a single, self-sufficient method for the specific and important model system considered in the manuscript. Instead, they were applied only in a minimum extent only in a handful of studies as shown in Table 1. Only two studies (ref 14, 17) employed LC-MS to study stability of lignin model compounds, which were structurally similar to Gb2; these studies either investigated different reaction pathways or used different model compounds. Therefore, this information clearly supports the conclusion that our study outcomes are novel.

As for the comment on peak shape, we did not need to make any additional  effort to further optimize the method, as the peak share of extracted ions is sufficient for effective separation, see Figure 3f. In Figures 3a-e, we show the total ion chromatograms, which convey changes in composition and give an overall idea of the results. If the editor deems this essential, we could revise this figure and show extracted ion chromatograms for all conditions.

We are surprised by the statement about the mass balance quality as most studies either do not report it at all or do not provide statistical variance. It appears that the rest of the Reviewers agrees with us. Reviewer 2 states: "The experimental evidence presented in the paper supports the study's objectives.” Similarly, the obtained separation is suitable to separate the feedstock from its decomposition products for the set objective. Reviewer 3 adds: "HPLC with high-resolution mass spectrometry was used to detect and quantify both Gβ2 and its degradation and modification products in an aqueous environment.”

The demonstration of advantages over GC is provided and stressed in the manuscript in the introduction section shown in the excerpt below. As for the comment on using HPLC as a single technique, with all due respect, the point of our work was the use of HPLC as a single technique for quantification of both the substrate and products.

The analysis of this literature (Table 1), however, reveals two significant knowledge gaps pertaining to the use of lignin model compounds. First, while a variety of product analysis methods were applied, only a few studies addressed the analysis of both reactant (e.g., Gβ2) and products using a single protocol [12-14]. Namely, most of the previous studies employed gas chromatography-mass spectrometry (GC-MS) for product characterization. This method enables a reliable identification and quantification of volatile reaction products, i.e., guaiacols and other methoxyphenol derivatives [14]. However, the direct GC-MS analysis does not allow for the simultaneous determination of reactants, i.e., hydroxylated dimer model compounds, most of which are either not GC-elutable, as is G2, or have low thermal stability [12-15]. The approaches to quantification was were thus restricted to the breakdown products in most of the conducted studies (Table 1), thus limiting the options in mass balance closure.

The above-mentioned papers that used a single GC-MS for determination of both Gβ2 and its decomposition products employed the derivatization of hydroxyl groups [12, 13]. However, this approach is not ideal for quantification, as the derivatization protocol must be optimized for any specific condition to assure its completion. Another approach is to quantify products directly by high-performance liquid chromatography with high resolution or tandem mass spectrometry (HPLC-HR MS or MS-MS) as reported by Rahimi et al [14].”

In summary,  we hope that we have clarified the issues, and we respectfully request that this review would not be considered to determine the fate of our manuscript. 

Reviewer 2 Report

Comments and Suggestions for Authors

This manuscript describes the efficiency of β-O-4 bond scission in guaiacylglycerol-β-guaiacyl ether (Gβ2), a lignin model compound. The study aims to address knowledge gaps in product analysis methods and the scarcity of baseline β-O-4 degradation data under moderate conditions in aqueous environments. The research highlights the limited applicability of Gβ2 for thermal hydrolytic β-O-4 bond scission at 150–250 °C due to significant side reactions. The study also evaluates Gβ2 stability at different pH, its biodegradation potential, and its breakdown in subcritical water.

The experimental evidence presented in the paper supports the study's objectives. I recommend this manuscript to be accepted by Separations.

Author Response

We are resubmitting our manuscript, addressing the comments of all the Reviewers. We would like to thank all reviewers for their feedback, as it helped us to improve all manuscripts and more clearly define the purpose of the manuscript, which was perhaps not fully transparent in our initial version. 

Thank you for this kind review.

Reviewer 3 Report

Comments and Suggestions for Authors

The manuscript “Stability and reactivity of guaiacylglycerol-β-guaiacyl ether, a compound modeling β-O-4 linkage in lignin“ by Z. Rabiei, A. Simons, M. Folkmanova, T. Vesela, O. Uhlik, E. Kozliak and A. Kubátová, describes how the title compound was used as a model to determine experimental conditions for the cleavage of the β-14-O-4 bond in lignin. A pH variation, microbial degradation using several strains, subcritical water conditions and mild pyrolysis were tried. HPLC with high resolution mass spectrometry was used to detect and quantify both Gβ2 and its degradation and modification products in an aqueous environment.

It was found that there were differences in Gβ2 breakdown under subcritical water treatment vs. pyrolysis under inert atmosphere, which were ascribed to the suppression of homolytic reactions when water was used as solvent.

The manuscript is fairly well written, although it still needs minor English revision. The references provided are adequate and recent, and the compounds were properly identified, although a question on this topic remains to be answered. Please see below.

The manuscript should be of interest to others in the field and I recommend it for publication in Separations, once the minor corrections listed below have been addressed.

Corrections required:

1. Quantitation: The word quantitative exists, but quantification is the form used, quantitation is not in the Oxford English dictionary. There are a few occurrences in the text, besides the word quantification.

2. P4, end of the first paragraph after scheme 2: “limiting the options in closing mass balance”; what does the word “closing” mean?

3. Table 1: In the heading “….summary of their degradation in aqueous media including the products obtained and their analysis protocols…” would be better as “….summary of their degradation data in aqueous media including the products obtained and the analysis protocols…”

4. Table 1 is interesting and informative, but the data should be presented in a concise manner; for example, for Depolymerization method & conditions, in entry 2 it appears …”6 bar H2”, but in entry 3 the pressure is reported as “P(H2) 40 bar,”. Please use the same format for everything. Check for other inconsistencies.

5. The explanation of all abbreviations should be given; i.e. OD = optical density?

6. Line 166: “stational phase” in “maximum OD600 and stational phase were determined;” Authors, do you mean “stationary phase”?

7. Table 2 should have column headings.

Please check the number of significant figures in the data in Table 2.

8. Line 236: quantitation => quantification

9. In scheme 3 and in scheme 4, the structure proposed for HMC seems strange. How can the Ph-CH3 bond be formed from Gb2? How can it be a degradation product?

10. And in HV and in CA??? Should this CH3 (in all three compounds) be a OH? Looking at the structure of D4 this seems likely too.

Please check.

12. Minor language corrections are needed.

13. References:

Commas are missing separating, year, issue, page numbers

Typos : ref 1 Page no/issue no., also ref 5, ref 6, please check for more.

Comments on the Quality of English Language

Minor language revision required.

Author Response

We are resubmitting our manuscript, addressing the comments of all the Reviewers. We would like to thank all reviewers for their feedback, as it helped us to improve all manuscripts and more clearly define the purpose of the manuscript, which was perhaps not fully transparent in our initial version. We provide responses below each review comment.

We have provided our responses and corrections below for each individual item.

Corrections required:

  1. Quantitation: The word quantitative exists, but quantification is the form used, quantitation is not in the Oxford English dictionary. There are a few occurrences in the text, besides the word quantification.

Response: Addressed as requested.

  1. P4, end of the first paragraph after scheme 2: “limiting the options in closing mass balance”; what does the word “closing” mean?

Response: Our goal is to achieve mass balance closure to clarify this we have changed “In closing mass balance” has been changed to “to determine mass balance closure.” We have verified that this specific term is used in applied chemistry and engineering.

  1. Table 1: In the heading “….summary of their degradation in aqueous media including the products obtained and their analysis protocols…” would be better as “….summary of their degradation data in aqueous media including the products obtained and the analysis protocols…”

Response: Corrected as requested.

  1. Table 1 is interesting and informative, but the data should be presented in a concise manner; for example, for Depolymerization method & conditions, in entry 2 it appears …”6 bar H2”, but in entry 3 the pressure is reported as “P(H2) 40 bar,”. Please use the same format for everything. Check for other inconsistencies.

Response: Addressed as requested. A couple of other inconsistencies have been fixed.

  1. The explanation of all abbreviations should be given; i.e. OD = optical density?

Response: The essential explanations have been provided in both Table 1 and Section 2.2.6. Line 166: “stational phase” in “maximum OD600 and stational phase were determined;” Authors, do you mean “stationary phase”?

Response: Thank you for pointing out to this typo, which has been corrected,

  1. Table 2 should have column headings.

Please check the number of significant figures in the data in Table 2.

Response: Thank you for noticing the inconsistency in the number of significant figures, which has been corrected. As for the column heading, they (e.g., “Experimental set of conditions 1”) appear to be redundant. We could add them if having column headings is required.

  1. Line 236: quantitation => quantification

Response: This word has been corrected throughout the manuscript.

  1. In scheme 3 and in scheme 4, the structure proposed for HMC seems strange. How can the Ph-CH3 bond be formed from Gb2? How can it be a degradation product?

Response: See the response to the next comment.

  1. And in HV and in CA??? Should this CH3 (in all three compounds) be a OH? Looking at the structure of D4 this seems likely too.

Please check.

Response: We are grateful to the Reviewer for finding this unfortunate typo, which has been corrected in both Schemes. It is indeed the hydroxyl group rather than methyl.

  1. Minor language corrections are needed.

Response: The manuscript has been checked for English quality with a native English speaker. Minor corrections have been made throughout the manuscript.

  1. References:

Commas are missing separating, year, issue, page numbers

Typos : ref 1 Page no/issue no., also ref 5, ref 6, please check for more.

Response: Thank you for pointing out to the problems created by the use of EndNote. The references have been checked and corrected manually to remove multiple formatting errors. DOI have been added, as required.

Reviewer 4 Report

Comments and Suggestions for Authors

The experimental article “Stability and reactivity of guaiacylglycerol-β-guaiacyl ether, a compound modeling β-O-4 linkage in lignin” is devoted to fundamental studies of the structure of native lignin and is a logical continuation (references 15 and 28, as well as a recent article by these authors in the publication Polymers (https://doi.org/10.3390/polym15193956) But in this case, the authors are armed with modern research methods (a high-performance liquid chromatography with high resolution mass spectrometry, gas chromatography-mass spectrometry analysis with a pyrolyzer ), study the stability and reactivity of guaiacylglycerol-β-guaiacylether (Gβ2) as a model substance. The experimental plan presented by the authors is quite bold and unpredictable, but the arguments they provide allow us to accept the conclusions drawn that the model substance Gβ2 retains its stability both when exposed to moderate pH levels, and several bacterial strains.Preliminary experiments showed that this substance should disintegrate. The scientific novelty of the article lies in the presentation of differences in the degradation pathways of Gβ2 during subcritical water treatment compared to pyrolysis in an inert atmosphere, with water playing a key role. The article provides enough information (in addition to drawings, chemical diagrams and Additional Materials are available) so that the reader can more thoroughly understand the scientific conclusions. But for its publication it is necessary to eliminate some comments.

Notes:

1. Introduction, first paragraph. It is necessary to provide a more modern link or even more than one for this statement, especially since there are currently no problems with this.

2. Materials and methods. It is necessary to supplement with a proposal about the reason for choosing these particular microorganisms for studying the model substance.

3. Section 2.2. Assessment of Gβ2 Stability. It is necessary to provide links to articles in which stability was studied in a similar way or to declare a pioneering author's approach.

4. Lines 290-302. The biological decomposition of the model substance did not occur in your studies. It is necessary to supplement this text with a sentence with a hypothesis about the cause of the observed phenomenon.

5. Scheme 4. Proposed Gβ2 moderate temperature pyrolysis breakdown pathways. The description of this scheme should be completed with a couple of sentences about how these results can be predicted for the decomposition of real lignin, thus connecting the discussion with the expression in the Introduction “...effectively studying lignin degradation pathways.” 

Author Response

We are resubmitting our manuscript, addressing the comments of all the Reviewers. We would like to thank all reviewers for their feedback, as it helped us to improve all manuscripts and more clearly define the purpose of the manuscript, which was perhaps not fully transparent in our initial version. 

We provide responses below each review comment.

  1. Introduction, first paragraph. It is necessary to provide a more modern link or even more than one for this statement, especially since there are currently no problems with this.

Response: A reference to an excellent, pertinent and comprehensive review published in 2023 has been provided to replace the old reference. Thank you for this comment that prompted us to include this reference.

  1. Materials and methods. It is necessary to supplement with a proposal about the reason for choosing these particular microorganisms for studying the model substance.

Response: The requested statement is there, at the end of Introduction, with references No. 18-20.

  1. Section 2.2. Assessment of Gβ2 Stability. It is necessary to provide links to articles in which stability was studied in a similar way or to declare a pioneering author's approach.

Response: The corresponding statements and references have been provided for 1) biodegradation; 2) subcritical water treatment and 3) mild pyrolysis.

  1. Lines 290-302. The biological decomposition of the model substance did not occur in your studies. It is necessary to supplement this text with a sentence with a hypothesis about the cause of the observed phenomenon.

Response: The requested concluding statement has been provided stating that “Gb2 turned out to be more recalcitrant toward biodegradation than anticipated. Therefore, it appears to be suitable as a model substrate to assess lignin biodegradation.”

  1. Scheme 4. Proposed Gβ2 moderate temperature pyrolysis breakdown pathways. The description of this scheme should be completed with a couple of sentences about how these results can be predicted for the decomposition of real lignin, thus connecting the discussion with the expression in the Introduction “...effectively studying lignin degradation pathways.” 

Response: We are particularly grateful to the Reviewer for pointing at this missed opportunity. The requested statement has been added at the end of Conclusion: “Extrapolating these results to lignin, the pathways of lignin depolymerization in subcritical water and as a result of solvent-free pyrolysis appear to have different nature, potentially leading to different products. The observed formation of the deoxygenated dimer as a result of Gb2 pyrolysis (as opposed to subcritical water treatment) indicates that repolymerization of intermediates of lignin breakdown in that system may occur more readily.”

Round 2

Reviewer 1 Report

Comments and Suggestions for Authors

In their response to reviewer, the author have fairly addressed my point. Therefore I would recommend the publication in separation.